# Learning to Mitigate AI Collusion on Economic Platforms

**Gianluca Brero**[*]
Data Science Initiative
Brown University
gianluca_brero@brown.edu

**Eric Mibuari**[*]
School of Engineering and Applied Sciences
Harvard University
mibuari@g.harvard.edu

**Nicolas Lepore**
School of Engineering and Applied Sciences
Harvard University
nlepore33@gmail.com

**David C. Parkes**
School of Engineering and Applied Sciences
Harvard University
parkes@g.harvard.edu

## Abstract

Algorithmic pricing on online e-commerce platforms raises the concern of tacit collusion, where reinforcement learning algorithms learn to set collusive prices in a decentralized manner and through nothing more than profit feedback. This raises the question as to whether collusive pricing can be prevented through the design of suitable "buy boxes," i.e., through the design of the rules that govern the elements of e-commerce sites that promote particular products and prices to consumers. In this paper, we demonstrate that reinforcement learning (RL) can also be used by platforms to learn buy box rules that are effective in preventing collusion by RL sellers. For this, we adopt the methodology of *Stackelberg POMDPs*, and demonstrate success in learning robust rules that continue to provide high consumer welfare together with sellers employing different behavior models or having out-of-distribution costs for goods.

## 1 Introduction

The last decade has witnessed a dramatic shift of trading from retailers to online e-commerce platforms such as Amazon and Alibaba. In these platforms, sellers are increasingly using algorithms to set prices. Algorithmic pricing can be beneficial for market efficiency, enabling sellers to quickly react to market changes and also in enabling price competition. At the same time, the U.S. Federal Trade Commission (FTC) U.S. Federal Trade Commission (2018) and European Commission (The Organisation for Economic Co-operation and Development, 2017) have raised concerns that algorithmic pricing may facilitate collusive behaviors. Calvano et al. (2020a) support these concerns through a study of pricing agents in a simulated platform economy, and show that commonly used reinforcement-learning (RL) algorithms learn to initiate and sustain collusive behaviors.[1] Assad et al. (2020) also provide empirical support for algorithmic collusion in a study of Germany's retail gas stations, showing an association between algorithmic pricing and an increase in price markups. As highlighted by Calvano et al. (2020b), these kinds of collusive behaviors are unlikely to be a violation of antitrust law, as they are learned responses to profit signals and not the result of explicit agreements.

One can try to prevent algorithmic collusion by introducing suitable rules by which platforms can choose which sellers to promote to buyers, thus promoting competition. Could Amazon's "buy box

---

[*]These authors contributed equally to this work.

[1]More precisely, Calvano et al. (2020a) showed that RL algorithms learn to quote supra-competitive prices and punish deviations from collusive agreements via lower prices.

algorithm," for example, play this role in the future, in determining for a given consumer search which products and prices to highlight to a consumer? Responding to this, Johnson et al. (2021) design hand-crafted rules that succeed in hindering collusion between RL algorithms. At the same time, their rules introduce the undesirable effect of limiting consumers to a single seller, and there remains potential for more effective interventions.

In this paper, we demonstrate for the first time how RL can also be used defensively by a platform to automatically design rules that promote consumer welfare and prevent collusive pricing. This is a problem of multi-agent learning, with the interaction between the platform and sellers modeled as a *Stackelberg game* (Fudenberg and Tirole, 1991). The leader is the platform designer and sets the platform rules and the sellers respond, using RL to set prices given these rules. We introduce the class of *threshold platform rules*, and formally show that this class contains rules that approximately maximize consumer surplus in a unique *subgame perfect equilibrium* (Mas-Colell et al., 1995, chapter 9). At the same time, this class of threshold rules is fragile to unexpected deviations by sellers, for example caused by cost perturbations. The role of RL on the part of the platform is to learn rules with similar performance that are also more robust.

To solve the Stackelberg problem, we make use of the *Stackelberg partially observable Markov decision process (POMDP)* framework (Brero et al., 2022), which defines an episode structure of a POMDP such that the RL algorithm representing the leader will learn to optimize its reward (here, consumer surplus) given that its rules cause re-equilibration on the part of the followers (here, the sellers who use Q-learning algorithms to set prices). The Stackelberg POMDP framework is well formed as long as the re-equilibration behavior of the sellers can be modeled through Markovian dynamics, as is the case with Q-learning.

We show successful results in learning effective platform policies that outperform handcrafted rules (Johnson et al., 2021). This demonstrates how the Stackelberg POMDP framework can be successfully applied in settings where followers play repeated games, and their strategies are also policies trained via reinforcement learning algorithms. We then show how our threshold platform rules allow us to obtain a similar learning performance when training the platform policy "in the wild," i.e., without accessing the sellers' private information. With this, we demonstrate how the Stackelberg POMDP framework can be applied in more general learning scenarios than the offline learning ones for which it was originally designed. Finally, we show how the platform rules learned via our Stackelberg POMDP framework continue to be effective when market conditions change, for example as the result of a change to the cost structure of sellers.

**Further related work.**  Zheng et al. (2022); Tang (2017); Shen et al. (2020); Brero et al. (2021) make use of RL to optimize different economic systems (including matching markets, internet advertising, tax policies, and auctions) under strategic agents' responses. Unlike our work, these methods do not leverage the designer's commitment power or the Stackelberg structure of the induced game. Brero et al. (2022) introduce and study the Stackelberg POMDP framework for matrix games and simple auctions using regret minimization for followers.[2]  Abada and Lambin (2022) study collusion by RL pricing in markets for electric power, and use machine learning by a regulator agent for the mitigation of collusion, albeit without a Stackelberg framing. The broader research program on *differentiable economics* uses representation learning for optimal economic design (Duetting et al., 2019; Shen et al., 2019; Kuo et al., 2020; Tacchetti et al., 2019; Rahme et al., 2021a; Curry et al., 2022; Rahme et al., 2021b; Curry et al., 2020; Peri et al., 2021); this work avoids the need for Stackelberg design by emphasizing the use of direct, incentive-compatible mechanisms. Also related is *empirical mechanism design* (Areyan Viqueira et al., 2019; Vorobeychik et al., 2006; Brinkman and Wellman, 2017), which applies empirical game theory to search for the equilibria of mechanisms with a set of candidate strategies (Wellman, 2006; Kiekintveld and Wellman, 2008; Jordan et al., 2010); see also Bünz et al. (2018) for the design of iterative auctions.

---

[2]The only other method we know for Stackelberg learning in stochastic games with provably guarantees solves for a single follower (Mishra et al., 2020); see also Mguni et al. (2019); Cheng et al. (2017); Shi et al. (2020) and Shu and Tian (2019), and Tharakunnel and Bhattacharyya (2007) for a partial convergence result for a static game with two followers. For other convergence results for single-follower, static, and often zero-sum games see Li et al. (2019); Sengupta and Kambhampati (2020); Xu et al. (2021); Fiez et al. (2020); Jin et al. (2020). For multi-follower static games, Wang et al. (2022) make use of a differentiable relaxation of follower best-response behavior together with a subroutine to solve an optimization problem for follower behavior.

## 2 Preliminaries

**Seller Competition Model.** There is a set of sellers $\mathcal{N} = \{1, \ldots, n\}$, each of whom sells a differentiated product on an economic platform. Each seller has the same *marginal cost* $c > 0$ for producing one unit of its product. Sellers interact with each other repeatedly over time in setting prices and selling goods. At each time period, $t = 0, 1, \ldots$, each seller $i$ observes all past prices, and sets a *price* $p_{i,t} \geq 0$. We let $p_t = (p_{1,t}, \ldots, p_{n,t})$ denote a generic price profile quoted at time $t$. The platform sets the rules of a buy box that governs, in each period $t$, which set $\mathcal{N}_t \subseteq \mathcal{N}$ of sellers are available. Consumers can only buy from these sellers and others forfeit sales. There is also an outside option, indexed by $0$, which provides each consumer with a fallback choice with zero utility.

Following Johnson et al. (2021), competition between sellers for consumer demand is modeled through the standard *logit model* of consumer choice. For this, seller $i$ has *quality index* $\alpha_i > 0$, this providing horizontal differentiation across products, and the outside good has quality index $\alpha_0 > 0$. In the logit model, each consumer samples $\zeta_0, \zeta_1, \ldots \zeta_n$, independently from a type I extreme value distribution with *scale parameter* $\mu > 0$, for each product and the outside option, with utility $\alpha_i + \zeta_i - p_{i,t}$ for product $i$, and $\alpha_0 + \zeta_0$ for the outside option. Considering a unit mass of consumers in period $t$, seller $i \in \mathcal{N}_t$ receives fractional demand $D_i(p_t; \mathcal{N}_t) = \exp((\alpha_i - p_{i,t})/\mu)/\lambda(p_t; \mathcal{N}_t)$, where $\lambda(p_t; \mathcal{N}_t) = \sum_{j \in \mathcal{N}_t} \exp((\alpha_j - p_{j,t})/\mu) + \exp(\alpha_0/\mu)$, and any seller $i \notin \mathcal{N}_t$ has zero demand. Scale parameter $\mu > 0$ serves to control the extent of horizontal differentiation, with no differentiation and perfect substitutes obtained as $\mu \to 0$. The total *consumer surplus* is $U(p_t; \mathcal{N}_t) = \mu \cdot \log[\lambda(p_t; \mathcal{N}_t)]$, and is maximized with minimum prices and all sellers displayed (so consumers have a full choice of products). Seller $i$'s *profit* $\rho_i$ in period $t$ is $\rho_i(p_t; \mathcal{N}_t) = (p_{i,t} - c) \cdot D_i(p_t; \mathcal{N}_t)$, i.e., its per-unit profit multiplied by demand.

**Reinforcement learning by sellers.** In a single-agent Markov decision process (MDP), an agent faces a sequential decision problem under uncertainty. At each step $t$, the agent observes a state variable $s_t \in S$ and chooses an action $a_t \in A$. Upon action $a_t$ in state $s_t$, the agent obtains reward $r(s_t, a_t)$, and the environment moves to state $s_{t+1}$ according to $q(s_{t+1}|s_t, a_t)$. We let $\tau = (s_0, a_0, \ldots, s_T, a_T)$ denote a state-action trajectory determined by executing policy policy $\pi : S \to A$, and $q_\pi(\tau)$ denote the probability of trajectory $\tau$. The optimal policy $\pi^*$ solves $\pi^* \in \text{argmax}_\pi E_{\tau \sim q_\pi(\tau)}[\sum_{t=0}^T \delta^t r(s_t, a_t)]$, where $\delta \in [0, 1]$ is the discount factor and time-horizon $T$ can be finite or infinite. In a *partially-observable MDP (POMDP)*, the policy $\pi$ cannot access state $s_t$ but only observation $o_t$ sampled from $q(o_t|s_t)$. A *multi-agent MDP* (Boutilier, 1996) for $n$ agents has states $S$ common to all agents and a set of actions $A_i$ for each agent $i$. When each agent $i$ picks action $a_{i,t}$ in state $s_t$, the environment moves to state $s_{t+1}$ according to a distribution $q(s_{t+1}|s_t, a_{1,t}, \ldots, a_{n,t})$ and agent $i$ obtains a reward $r_i(s_t, a_t)$ that depends on the joint action. We follow Calvano et al. (2020a) and Johnson et al. (2021) and adopt decentralized Q-learning by sellers as a positive theory for sellers in regard to their behavior in setting prices on an e-commerce platform (see Appendix A). Although Q-learners may not converge, we also confirm these earlier studies in showing convergence in our simulations (defined over a particular time horizon as detailed by Johnson et al. (2021)).

## 3 The Platform Stackelberg Problem

To formalize the problem facing the platform designer in mitigating collusive behavior by sellers, we model the interaction between the platform, which sets the rules of the buy box, and the sellers as a *Stackelberg game*. The platform designer is the leader, and fixes the platform rules. The sellers are the followers, and play an infinitely repeated game according to these rules. As discussed above, and following Calvano et al. (2020a) and Johnson et al. (2021), we model the sellers' behavior through decentralized Q-learning. As a result, the problem facing the platform is a *behavioral Stackelberg problem*, in that the followers are modeled as Q-learners (and need not, necessarily, be playing an equilibrium of the induced game).

**The sellers.** In this model, we fix the states that comprise the MDP of a seller to include the prices set by all sellers in the last period, i.e., $s_t = p_{t-1}$. We initialize $s_0$ to be a randomly selected price profile. The action of a seller is modeled as one of $m$ equally-spaced points in the interval ranging from just below the sellers' cost $c$ to just above the monopoly price $p^m$. At each step $t \geq 0$, each

seller $i$ selects a price $p_{i,t}$ and is rewarded by its per-period profit $\rho_i(p_t; \mathcal{N}_t)$, which depends on $p_t = (p_{1,t}, \ldots, p_{n,t})$ and the choice of which sellers $\mathcal{N}_t$ are displayed by the platform.

**The platform.** To formalize the platform's problem, let $\sigma^* = (\sigma_1^*, .., \sigma_n^*)$ denote a strategy profile selected by Q-learning on the part of sellers, in response to the platform rule, and in the long run, after a suitably large number of steps. We leave implicit here the dependence of seller strategy profile on the platform's policy. The platform must decide in each period which sellers to display to consumers. For this, we denote the platform rule as *policy* $\pi$, and we adopt for the state of the platform policy the prices quoted by sellers in step $t$, $p_t$, so that the platform's policy uses these prices to select a set of agents to display, with $\mathcal{N}_t$ selected according to $\pi(p_t)$. Let $p_t^* = \sigma^*(s_t)$ denote a price profile chosen under seller strategies $\sigma^*$, i.e., in response to the platform rules, and at some large enough time step $t^*$, and let $\tau^* = (p_{t^*}^*, p_{t^*+1}^*, ..)$ denote a trajectory of prices forward from $t^*$ (the dependence on the platform's policy is left implicit in this notation). We denote the distribution of these trajectories as $q_\pi(\tau^*)$. The Stackelberg problem facing the platform is to find a platform policy $\pi$ that maximizes consumer surplus given the effect of this policy on the induced strategy profile of sellers.

**Definition 1 (Behavioral Stackelberg Problem)** *The optimal platform policy solves* $\pi^* \in argmax_\pi CS(\pi)$, *where* $CS(\pi)$ *is the expected sum consumer surplus when sellers follow strategy* $\sigma^*$ *forward from period* $t^*$, *i.e.,*

$$CS(\pi) = \mathbb{E}_{\tau^* \sim q_\pi(\tau^*)} \left[ \sum_{t=t^*}^{T^*} U(p_t^*; \pi(p_t^*)) \right], \tag{1}$$

*where* $T^*$ *is a suitably chosen horizon[3] and* $q_\pi(\tau^*)$ *denotes the distribution over Q-learning induced seller pricing trajectories in response to platform policy* $\pi$.

## 4 Learning Optimal Platform Rules

In this section, we solve the platform's problem, in responding to Q-learning sellers, through the *Stackelberg POMDP* framework (Brero et al., 2022). This creates a suitably defined POMDP in which the optimal policy solves the behavioral Stackelberg problem (Definition 1).

**Definition 2 (Stackelberg POMDP for platform rules)** *The* Stackelberg POMDP *for platform rules is a finite-horizon POMDP, where each episode has the following two phases:*

• *An* equilibrium phase, *consisting of* $n_e \geq 1$ *steps. In this phase, each state* $s_t$ *includes the step counter* $t$, *the sellers' current Q-matrices, and the prices* $p_t$ *quoted by the agents. Observations consists of the prices quoted by the sellers* ($o_t = p_t$) *and policy actions determine the set of agents displayed (in their more general version,* $a_t = \mathcal{N}_t$). *State transitions are determined by Q-learning, where each seller* $i$ *updates its Q-matrix after being rewarded by* $\rho_i(p_t; \mathcal{N}_t)$. *The policy has zero reward in every time step* ($r(s_t, a_t) = 0$, *for* $t \leq n_e$).

• *A* reward phase, *consisting of* $n_r \geq 1$ *steps, each with the same actions, states, and observations as the equilibrium steps. The reward phase differs in two ways. First, the Q-matrices of the sellers are not updated, and second, the platform policy now receives a non-zero reward, and this is set in each step to be equal to the consumer surplus in that step* ($r(s_t, a_t) = U(p_t; \mathcal{N}_t)$, *for* $t > n_e$).

This Stackelberg POMDP formulation is an adaptation of that provided by Brero et al. (2022), who used it to learn leader strategies in matrix games and allocation mechanisms. Following Brero et al. (2022), we show the Stackelberg POMDP formulation is well-founded by showing that an optimal policy will also solve the Behavioral Stackelberg design problem of Definition 1. Specifically, when the number of reward steps $n_r$ is large enough and when $n_e \geq t^*$, the optimal policy, denoted $\pi_{n_e, n_r}^*$, for the Stackelberg POMDP with $n_e$ equilibrium and $n_r$ reward steps maximizes the objective in Equation (1).

**Proposition 1** *The optimal Stackelberg POMDP policy* $\pi_{n_e, n_r}^*$, *for an equilibrium phase with* $n_e \geq 1$ *steps and a reward phase with* $n_r \geq 1$ *steps, maximizes* $CS(\pi)$, *for seller behavior induced after* $n_e$ *steps when* $n_r = T^*$.

---

[3]Note that we can use a finite time horizon as trajectories $\tau^*$ consists of price cycles due to our sellers' behavior model.

The proposition follows from the construction of the Stackelberg POMDP, especially the fact the our policy is only rewarded under the response behavior reached after $n_e$ steps, in line with the definition of $CS(\pi)$ (see Appendix B for the full proof argument).

Brero et al. (2022) use the Stackelberg POMDP framework in an "offline" environment, i.e., in a simulation that assumes access, at design time, to followers' internal information. This allows them to solve their POMDP using the paradigm of *centralized training and decentralized execution* (Lowe et al., 2017). The leader policy is trained via an actor-critic deep RL algorithm, and the critic network (which estimates the sum of rewards until the end of the episode) accesses the full state during training. Only the actor network, which represents the policy, is restricted to the partial-state information.

Here, we also study the use of the Stackelberg POMDP framework to train useful leader policies "in the wild," where the learning algorithm of the platform can only access the kind of information that an economic platform would have based on observations of sellers. As we will empirically demonstrate, we can successfully operate without access to sellers' private information in regard to Q-matrices and exploration rate without affecting learning performance.[4]

**Threshold platform rules.** In our experiments, we consider the class of *threshold platform rules*. These rules use the current prices set by sellers to set a price threshold above which a seller will not be displayed, with the same threshold set for all sellers.

**Definition 3 (Threshold Platform Rule)** *A threshold platform rule sets a threshold $\tau(p_t) \geq 0$, for each price profile $p_t$, such that $\mathcal{N}_t = \{i \in \{1, .., n\} : p_{i,t} \leq \tau(p_t)\}$, i.e., any seller whose price is no greater than the threshold is displayed to consumers.*

This class of threshold rules has a corresponding optimality result: there is a threshold rule that makes the market competitive, with all sellers displayed and consumer surplus maximized in a subgame perfect Nash equilibrium (SPE) of the induced continuous pricing game. Even though the pricing behaviors that arise from Q-learning need not converge to SPEs, we use these equilibria as a theoretical support for our platform intervention choice, as previously done by Johnson et al. (2021). We have the following result:

**Proposition 2** *For any $\epsilon > 0$, there exists a threshold platform rule $\pi$ such that $CS(\pi) \geq CS(\pi^*) - \sum_t \delta^t \epsilon$ under a subgame perfect Nash equilibrium (SPE) of the infinitely-repeated continuous pricing game induced by platform rule $\pi$.*

This proposition follows from a platform rule with a limiting threshold that is arbitrarily close to the sellers' cost $c$ (see Appendix C for the proof). Under this rule, sellers are displayed only if their price is minimal. At the same time, this particular threshold rule is fragile, and would lead to market failure if seller costs vary in unexpected ways. By letting the threshold $\tau$ also vary with the price profile $p_t$, as is allowed by the family of threshold platform rules, we seek to learn milder interventions that still mitigate collusion but remain robust to variations in the costs faced by sellers in the marketplace.

# 5   Experimental Results

In this section, we evaluate our learning approach via three main experiments. We first consider performance in terms of consumer surplus, benchmarking our RL interventions against the ones introduced by Johnson et al. (2021). We demonstrate the ability to learn optimal leader strategies in the Stackelberg game with the followers across all the seeds we tested, significantly outperforming existing interventions. We then train platform rules without access to the sellers' private information ("in the wild"), and show that this is not crucial for our learning performances. We conclude by testing the robustness of our interventions, adding price perturbations during training and evaluating the effect on the robustness of our learned platform rules in environments where sellers have different costs from those assumed during training.

---

[4]We notice this is also in line with the recent findings in Fujimoto et al. (2022) highlighting how the Bellman error minimization (for which we require environments to be Markovian) may not be a good proxy of the accuracy of the value function.

**Experimental set-up.** As in Calvano et al. (2020a) and Johnson et al. (2021), we consider settings with two pricing agents with cost $c = 1$, quality indexes $\alpha_1 = \alpha_2 = 2$, and $\alpha_0 = 0$, and we set parameter $\mu = 0.25$ to control horizontal differentiation. The seller Q-learning algorithms are also trained using discount factor $\delta = 0.95$, exploration rate $\varepsilon_t = e^{-\beta t}$ with $\beta = 1e - 5$, and learning rate $\alpha = 0.15$. We also include results for variations of this default setting in Appendix D. A particular concern is that if the exploration rate is still high during the reward phase of the Stackelberg POMDP episode, our policy may be rewarded based on random prices. To address this problem, one could extend the Stackelberg POMDP equilibrium phase to reach a minimal $\varepsilon_t$ or isolate stages when price profiles are more stable to audit rewards. To satisfy our computational constraints, we pause exploration during the reward phase of the Stackelberg POMDP. As in Johnson et al. (2021), these prices range from 0.95 (just below the sellers' cost) to 2.1 (which is above the monopoly price under no intervention). Earlier work provided sellers with a choice of fifteen different prices (over a similar range). We need a smaller grid in order to satisfy our computational constraints; earlier work studied the effect of different, hand-designed platform rules, and did not also use RL for the automated design of suitable platform rules. We also follow the choices of earlier work, and study an economy with two sellers (again, for reasons of computational resources). This coarsened price grid allows us to train a platform policy through Stackelberg POMDP for 50 million steps in 18 hours using a single core on a Intel(R) Xeon(R) Platinum 8268 CPU @ 2.90GHz machine.

**Learning algorithm.** To train the platform policy, we start from the A2C algorithm provided by *Stable Baselines3* (Raffin et al., 2021, MIT License). Given that our policy is only rewarded at the end of a Stackelberg POMDP episode, we configure A2C so that neural network parameters are only updated after this reward phase. In this way, we guarantee that policies inducing desired followers' equilibria are properly rewarded. Furthermore, to reduce variance in sellers' responses due to non-deterministic policy behavior, we maintain an observation-action map throughout each episode. When a new observation is encountered during the episode, the policy chooses an action following the default training behavior and stores this new observation-action pair in the map. We will show the importance of this variation via an ablation study that is presented in Appendix E. In this section, we assume that sellers restart the Q-learning process by re-initializing exploration rates every time the platform rules change (i.e., at the beginning of every Stackelberg POMDP episode). In Appendix F, we also show how the training approach is robust to different sellers' behaviors, where the sellers restart the learning rate asynchronously, and not necessarily at the beginning of episodes.

## 5.1 Platform Learning Performance

In this section, we evaluate the performance of our learned platform policies. For this, we train our policies for 50 million steps in total. We set up the Stackelberg POMDP environment using 50k equilibrium steps and 30 reward steps.[5] In these initial experiments, we train our policies using the centralized training-decentralized execution paradigm as used for this Stackelberg learning problem by Brero et al. (2022), giving the critic network access to the sellers' learning information (i.e., Q-tables and exploration rates). We relax this below in studying the robustness of the computational framework to online training ("in the wild"). We consider the following interventions on behalf of the platform designer:

• *No intervention:* Sellers are always displayed, no matter the price they quote. To derive this baseline, we run our Q learning dynamics until convergence (as described in Johnson et al. (2021)) for each seed and then average the surplus at final strategies.

• *PDP:* We test *price-directed prominence*, a platform intervention introduced by Johnson et al. (2021). Here, the platform only displays the seller who quotes the lower price (breaking ties at random), thus enhancing competition. As for *no intervention*, we compute the performance of *PDP* by averaging consumer surplus after Q-learning dynamics converge.

• *DPDP:* Dynamic price-directed prominence is another intervention introduced by Johnson et al. (2021), which also conditions the choice of the (unique) displayed seller on past prices. Under this intervention, quoting prices equal to cost is a subgame perfect equilibrium of the induced game (under suitable discount factors). As for the previous baselines, we compute the performance of *DPDP* by averaging consumer surplus after Q-learning dynamics converge.

---

[5]See Appendix H for a discussion around parameter selection.

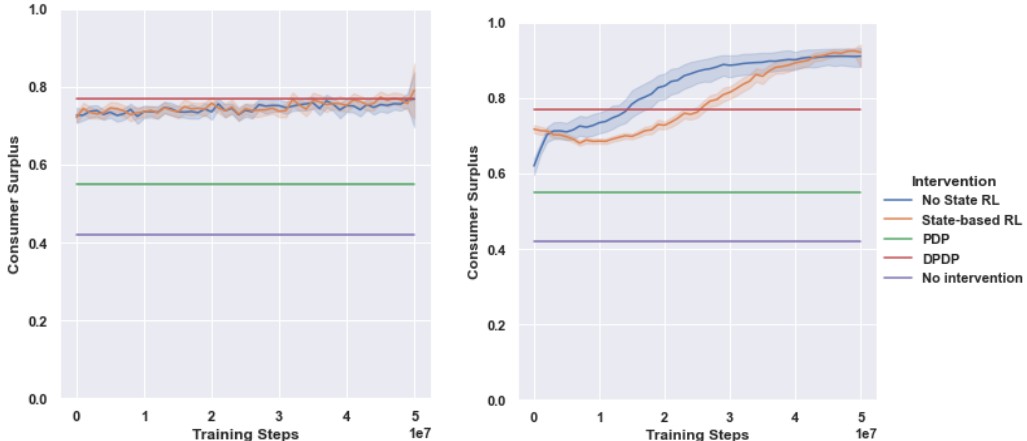

Figure 1: Learning performance of No State RL and State-based RL compared with different baselines. The results are averaged over 50 runs and shaded regions show 95% conf. intervals. The No-Stackelberg interventions are displayed on the left, the Stackelberg ones are on the right.

- *No State RL:* Here we use the Stackelberg POMDP methodology to train a platform policy that does not use prices $p_t$ to determine the threshold at which to admit each seller to the buy box (thus, "no state").[6] Here, Q-learning is restarted whenever a Stackelberg POMDP episode begins.

- *No Stackelberg No State RL:* A variation on "No State RL" that does not use the Stackelberg POMDP methodology. Rather, the platform and sellers each follow decentralized learning, and the platform receives a consumer surplus reward at every step. Q-learning is restarted after the same number of steps that are used in a Stackelberg POMDP episode.

- *State-based RL:* Here we use the Stackelberg POMDP methodology to train a platform policy that sets a threshold at which to admit each seller as a function of the price profile quoted by the sellers (thus, "state-based"). This is the full class of threshold platform rules. Here, Q-learning is restarted whenever a Stackelberg POMDP episodes begins.

- *No Stackelberg State-based RL:* A variation on "State-based RL" that does not use the Stackelberg POMDP methodology. Rather, the platform and sellers each follow decentralized learning, and the platform receives a consumer surplus reward at every step. Q-learning is restarted after the same number of steps that are used in a Stackelberg POMDP episode.

Figure 1 shows the consumer surplus that is realized under these different interventions. First, we confirm the results of Johnson et al. (2021), and see consumer surplus improvements from both *PDP* and *DPDP* compared to *No intervention*, with *DPDP* outperforming *PDP*. At the same time, the no Stackelberg baselines are not able to outperform DPDP, confirming the benefits of using learning methodologies that exploit the leader-follower structure of our game. Indeed, our RL interventions based on the Stackelberg framework dramatically improve consumer surplus, driving it to (approximately, in the state-based scenario) its maximal level. In our setting, this optimal level for surplus is approximately 0.94. This is confirmed by the fact that, for both *No State* and *State-based* RL, all sellers are displayed and they invariably quote minimum prices at the end of training. This is the optimal (i.e., surplus maximizing) seller behavior, confirming the effectiveness of the Stackelberg-based learning methodology in finding an optimal leader strategy given the Q-learning behavior of sellers. It is easier for *No State RL* to reach the optimal performance since its class of policies is much smaller than the class considered by *State-based RL*. However, as we will see in Section 5.3, the state-based policy is more flexible and is robust to the case that the cost basis changes for sellers while *No State RL* is not.

---

[6]This class of policies already includes the optimal policy described in the proof of Proposition 2.

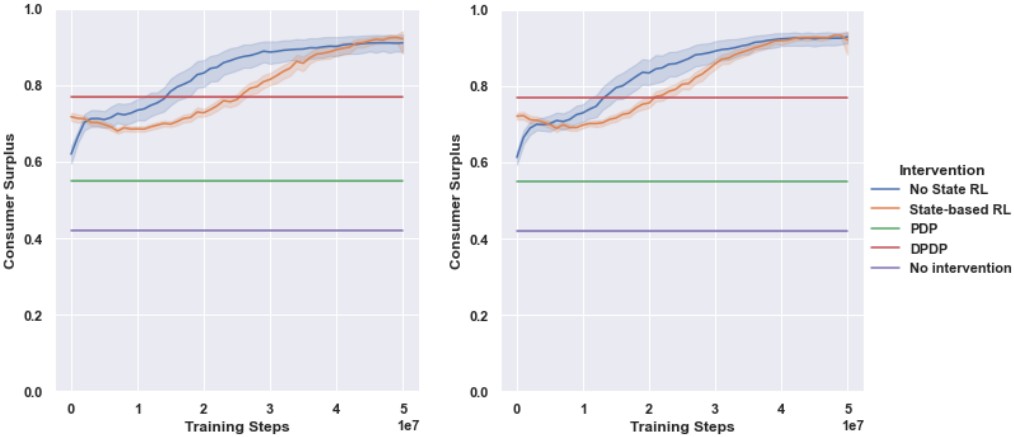

Figure 2: Offline learning (left) vs. online, "learning in the wild" (right) performance. The results are averaged over 50 runs, and the shaded regions show 95% conf. intervals.

## 5.2 Learning in the Wild

We now test the performance of the Stackelberg POMDP learning methodology when it has no access to sellers' private information during training. This can potentially create learning instabilities given that actor-critic training such as A2C generally require that the environment accessed by their critic networks is Markovian (Grondman et al., 2012). Despite this, we find success in this test of "in the wild" learning. The results are displayed in Figure 2 and show, despite relaxing this Markov assumption, that the A2C algorithm is able to learn optimal policies for both policy classes (No State and State-based). We conjecture that the reason behind this good performance is related to the class of threshold platform policies. Given a threshold policy, it is possible to predict the overall episode reward based only on the action taken by the policy (the threshold) and ignoring the part of the state that is internal to the sellers (i.e., the Q-matrix and exploration rate).

In Appendix G, we also demonstrate successful experimental results when we replace the use of consumer surplus (1) for reward with a reward that corresponds to the number of agents displayed and the sum of the negated prices offered by sellers. This shows robustness to a possible knowledge gap in knowing the specific functional form of consumer surplus.

## 5.3 Robustness of Learned Platform Rules

As observed in our previous experiments, the Stackelberg-based RL algorithm is effective in learning interventions that maximize consumer surplus for a given economic setting. However, as they are tailored to the economic setting at hand, these interventions can perform poorly when facing settings that are different from those during training. To learn more robust platform rules, we also train with a modified version of the Stackelberg POMDP: at each reward step, with some *random-price probability*, sellers quote prices sampled uniformly at random from the price grid. In this way, the platform is rewarded during training for performance that remains robust to prices that are not produced by the Q-learning equilibrium dynamics (given seller costs at training).

We evaluate the effect of adding this perturbation-based robustness to the training procedure in settings with different seller costs: in addition to the default $c = 1.0$, we also test with cost $c = 1.38$ (between the second and the third price in the grid of prices between 0.95 and 2.1) and cost $c = 1.67$ (between the third and the fourth price in the price grid).

As we see in Figure 3 (right), this training approach (and in particular with probability 0.4 of random-price perturbation) succeeds in making the state-based policy much more robust in the face of sellers who experience a different cost environment at test time. The robust, state-based policy displays sellers with higher prices (due to their higher costs), while continuing to significantly mitigate collusion when seller costs are as they were during training. This is also confirmed by the policy visualizations in Figure 3 (left), which show how the buy box learned for State-based RL tends to be

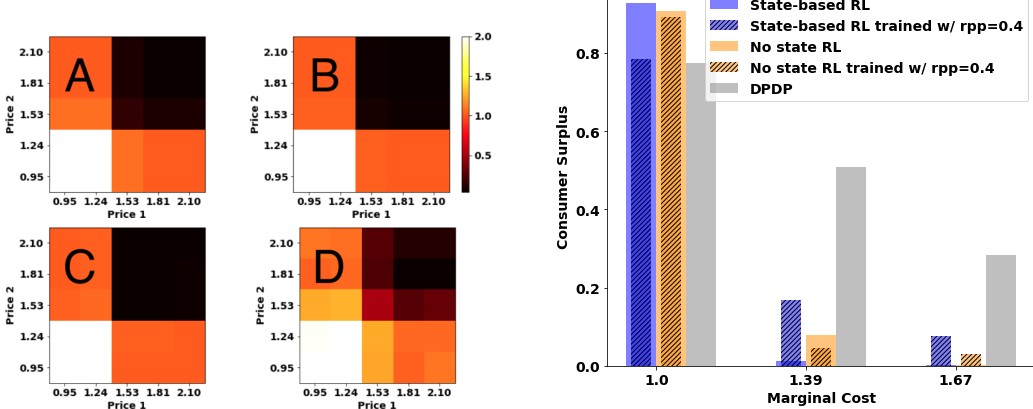

Figure 3: Left: **Policy visualization**, with number of displayed agents given price selection, averaged over 50 seeds (white–avg. num. sellers displayed 2, black–avg. num. sellers displayed 0). A: No State RL with no price perturbation during training. B: No State RL with 40% random price perturbation (rpp) during training. C: State-based RL with no price perturbation during training. D: State-based RL with 40% random price perturbation during training. Right: **Robustness test**, with buy box policy trained without price perturbation and with price perturbation with prob. 0.4, averaged over 50 runs.

much more open under this modified training regime. In contrast, the policy learned by No State RL performs very poorly when tested at costs that differ from those assumed during training, and even under this modified training regime. There is no single threshold that provides a good compromise between performance at cost 1 and handling price perturbations.

# 6   Conclusion

This work has demonstrated that rules that are effective in preventing collusion by sellers can be learned through a framework that correctly solves the two-level, Stackelberg problem (making use of the platform's commitment power). Specifically, we have introduced the class of *threshold policies* that contain policies that optimize consumer surplus and a learning methodology that is effective in learning optimal leader policies in this class. The interventions we learned are shown to substantially outperform the hand-designed interventions introduced in prior work when the cost environment at test time is as anticipated during training. We also showed how our learned platform interventions can be made more robust when settings are dynamic, with varying seller cost structures, by adopting a suitably-modified training methodology. This also highlights the importance of the state-based platform rule relative to a no-state rule.

Interesting future directions include testing our approach in more complex settings, e.g., when sellers' costs vary between training episodes. In this case, optimal policy actions are based on the prices quoted during the sellers' equilibration phase, as these prices may provide useful information about the current underlying costs (intuitively, the quoted prices will be higher under higher costs). In this scenario, it may be necessary to represent our platform policies via recurrent neural networks, keeping a memory of past prices. Finally, we believe that this approach can also be effective in other applications, e.g., to design and understand effective interventions for the electricity markets studied by Abada and Lambin (2022), a setting where the successful use of RL as a defensive response by a platform is not yet established.

# Acknowledgements

This research is funded in part by Defense Advanced Research Projects Agency under Cooperative Agreement HR00111920029. The content of the information does not necessarily reflect the position or the policy of the Government, and no official endorsement should be inferred. This is approved for public release; distribution is unlimited. The work of G. Brero was also supported by the SNSF

(Swiss National Science Foundation) under Fellowship P2ZHP1_191253. We thank Emilio Calvano and Justin Johnson for their availability to answer questions about their work and for guidance in replicating some of their results. We also thank Alon Eden, Matthias Gerstgrasser, and Alexander MacKay for for helpful discussions and feedback. Many computations in this paper were run on the FASRC Cannon cluster supported by the FAS Division of Science Research Computing Group at Harvard University.

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
