# OpenReview forum: "Learning to Mitigate AI Collusion on Economic Platforms"
_NeurIPS.cc/2022/Conference — NeurIPS 2022 Accept_

### Official Review · Reviewer_DPgu · 2022-07-09

**Rating:** 6
**Confidence:** 4
**Soundness:** 3 good
**Presentation:** 3 good
**Contribution:** 3 good

**Summary:**

This paper uses reinforcement learning (RL) by a platform to learn buy box rules that are effective in preventing collusion by RL sellers. The authors make use of the approach of Stackelberg POMDPs.

**Questions:**

Have the authors performed additional simulations? Do different simulations provide further insights or further demonstrate competitiveness of your proposed approach?

**Strengths And Weaknesses:**

The strengths of the paper are that the topic is an interesting and important one, and the proposed approach is competitive in reducing collusion on platforms under the studied setting.

The main weakness of the paper is that the numerical simulations, including the ones in the Appendix, are not very comprehensive. It would have been interesting to see numerical simulations in more different settings than the manuscript had. An additional weakness is that the paper does not provide strong or interesting theoretical characterizations in the applicable settings.

---

> ### Author Response · Authors · 2022-08-01
> **Reviewer DPgu Rebuttal**
>
> Thank you for your thorough feedback!
>
> We have included new results in Figure 4 of Appendix C. These results are for a setting that has greater horizontal differentiation across products (\mu=0.4) and show that, even in this scenario, we can learn optimal interventions.
>
> On studying a wider variety of economic environments: we have deliberately chosen to stick close to the model introduced by Calvano et al. (2020), which has become the benchmark model for algorithmic collusion by pricing agents on economic platforms and is the same model studied in follow-on work by Johnson et al. (2020). This makes our results on the use of RL for platform rule design directly comparable to the existing economics literature. Future extensions to other models will first require new studies around AI collusion. This is an interesting research direction, but one that we feel is more in scope for a future paper.

---

### Official Review · Reviewer_aSeB · 2022-07-15

**Rating:** 5
**Confidence:** 4
**Soundness:** 3 good
**Presentation:** 2 fair
**Contribution:** 2 fair

**Summary:**

This paper proposes a reinforcement learning based method to prevent the collusive pricing,
which is important to the development of AI based economic platforms.

Specifically, they model the economic platforms as a Stackelberg where the leader is the platform and the followers are the sellers.
The leader uses the AC to train while the follower is Q learning based.

The authors provide some interesting theoretical results (Although some lacks important steps).

The experiments show that the proposed method is well-performing comparing with some baselines.

**Questions:**

Please answer the questions mentioned above.

**Limitations:**

It would be a plus if the authors could provide more details on 1. mentioned in 'Strengths And Weaknesses' part.


Since this paper focuses on economic system, the potential negative societal impact lies in how the learned platforms are stable enough. The authors propose a robust analysis, which address my concerns.

**Strengths And Weaknesses:**

Strengths:
The proposed method is robust and the ablations are detailed.


Weaknesses:
1. This paper is limited in solving a special economic platform.
That is, the paper (including both the theoretical and experimental studies) is mainly focusing on
those systems that can be represented in lines 88-100, and it is doubtful that whether this type of economic platform is general enough.

Could the authors discuss more details about whether this economic platform is widely adopted and used?

2. Novelty is somewhat minor, the used RL methods have already been introduced in Mguni et al. (2019); Shi et al.
(2020) and Shu and Tian (2019).
The proposed threshold platform rule is new to me, but it only fits a special economic platform.

3. Regarding the empirical studies, the baselines seem not strong enough. The authors discuss many related RL methods to solve Stackelberg game but does not compare them in the experiments. Is it possible to compare some of them?

Minors:

1). The notations are somewhat ambiguous. The authors firstly define $p$ as generic price profile, but then reload it as probability, which incurs some ambiguity. It would be better to change the notation of the generic price profile to another.

2). some important definitions are missing, e.g., Nash equilibrium. It would be better to define them.

3). Some of the proofs are lacking important steps, e.g., in lines 592-596 (Prop. 2), why $\rho(p;\mathcal{N})$ is strictly increasing implies Nash equilibrium. To my best knowledge, Nash equilibrium is a set of mixed strategies from which no player has a unilateral
incentive to deviate [1]. It would be nice to add more details.

[1] Daskalakis, Constantinos, Paul W. Goldberg, and Christos H. Papadimitriou. "The complexity of computing a Nash equilibrium." SIAM Journal on Computing 39.1 (2009): 195-259.



####post-rebuttal####

Thank the authors for the feedback. It helps the reviewer better understand the proposed method.

i) However, the reviewer is still somewhat concerned about the limited scope of this paper, as the proposed algorithm is only designed and testified in a specific economic model. Reviewer DPgu and Reviewer VP8z also ask similar questions.

Although the authors provide some related works [Calvano et al. (2020a), Johnson et al. (2020)],  they all belong to the economic area. For the submission to an AI-related conference, it would be nice to be more general.

ii) Also, regarding the Nash Eq analysis, it is recommended to add additional definition and proof directly to the revision, rather than just roughly saying the proof sketches, as this hand-waving claim is hard for the reviewer to check whether it is correct.

The reviewer is also confused about why the authors promised to add proofs and formal definitions but ignored them when they submitted the revision.

---

> ### Author Response · Authors · 2022-08-01
> **Reviewer aSeB Rebuttal**
>
> Thank you for your thorough feedback!
>
> Weaknesses1: We chose to investigate the setting studied by Calvano et al. (2020), which has become the benchmark model in economics and sets up Q-learning as a positive theory for the behavior of AI agents in this kind of economic environment.  This same model was then used by Johnson et al. (2020) to study the use of platform rules to mitigate price collusion. For this reason, we prefer to stick with this earlier modeling decision because it provides a clear comparison to non-automated methods, and we leave for future work also adopting alternate models of seller behavior.
>
> Weaknesses2 (novelty): We build on the StackPOMDP framework introduced in Brero et al. (2021a). We differentiate from them in that we learn platform interventions instead of auction rules, our followers are using RL algorithms instead of no-regret ones, and our policies do not access the sellers’ internal information at training time. The RL methods you mention are very different from StackPOMDP: Mguni et al. (2019) introduce black-box optimization algorithms to modify rewards in Markov potential games (different from that studied here), and Shu and Tian (2019) and Shi et al. (2020) study settings where the leader interacts with followers by assigning them bonuses and tasks. These methods cannot be used in the present application,  as we want to address the specific, application-motivated problem of learning buy-box policies that observe quoted prices and determine subsets of sellers to display to buyers.
>
> Weaknesses3 (baselines): please see our answer to Weaknesses2. Note also that our NoState RL can be interpreted as a multi-armed bandits approach as we select a policy from among five arms by reinforcing weights via rewards. As we show, despite being competitive under fixed costs, this can lead to major failures for out-of-distribution costs. Indeed, with only five thresholds, the policy cannot prevent collusion while giving enough leeway to sellers, for example in case their cost basis changes. If we let the threshold depend on quoted prices, the number of platform policies grows exponentially, and bandit approaches become unfeasible. Even when we only let the threshold depend on the last quoted prices (as in our state-based RL), the number of platform policies (= arms) is 5^{25}. Thus, RL methods become necessary.
>
> Regarding minor comments: Thanks for catching the notation problem! We will fix this in our revision and define Nash eq. as you suggested.
>
> On the Nash eq analysis: as \rho_i(p_i, p_{-i}; N) is strictly increasing, agent i can always increase its utility (which creates an incentive to deviate from p_i) by quoting p_i+eps instead of p_i. This happens unless p_i is equal to c+\eta: in this case, increasing p_i would drive the agent out of the buybox (zeroing its utility). Thus, there cannot be equilibria where an agent sets a price different from c+\eta. We will add a comment to make sure this is clear.

---

### Official Review · Reviewer_7ViN · 2022-07-21

**Rating:** 7
**Confidence:** 2
**Soundness:** 3 good
**Presentation:** 2 fair
**Contribution:** 3 good

**Summary:**

This is an emergency review of the paper.

A recent topic on the border of ML and economics is that of reinforcement learning approaches that learn "tacit collusion" in the sense that decentralized independent self-play based on these algorithms in various auction settings reaches non-competitive (collusive) prices.

The authors take a stab at tackling this problem by designing a mechanisms for ecommerce sites that promotes produces and prices to consumers so as to prevent such tacit RL collusion (so-called applying a suitable "buy-box"). The authors show that certain such buy-box rules indeed manage to avoid such tacit collution, at least for various types of RL methods. The methodology is based on a leader-follower model (Stackelberg) on a reasonable POMDP environment. Hence, the authors show that such rules manage to keep consumer welfare high.

**Questions:**

See above - can you consider additional RL algorithms (there are lots of libraries you can just launch on this environment, I think). Also, any thoughts on how the welfare is distributed among the sellers?

**Limitations:**

The authors acknowledge the limitations here, but I think a more detailed discussion of various RL algorithms is warranted.

**Strengths And Weaknesses:**

Strength: I really love the topic of the paper. Tacit collusion is indeed an important problem in the age of algorithmic trading, and due to the immense progress in RL algorithms (especially multi-agent deep RL), it makes sense to see whether bad behaviours can be prevented. This topic is very recent, and should, in my opinion get more attention.

I find the model reasonable, and the suggested mechanism simple yet effective.

The weakeness of the paper lies in its presentation. First, I think the results may be different depending on the RL approach used. You consider both Q-based approaches and policy gradient / actor critic approaches. However, I would have liked a more detailed experimental analysis, considering multiple algorithms, from vanilla reinforce, SARSA (tabular and using a neural function approximator) as well as DQN. You also examine A2C, but I think it makes sense to also examine other algorithms (V)MPO and PPO, for example. Are the results sensitive to choices of hyperparameters (e.g. learning rates, neural architecture, optimizer)? Please be clearer about where your theoretical results hold (e.g. in the form of a table, early in the paper), and where you have empirical analysis supporting the claims. If the results are sensitive to the type of RL applied, any inisght as to the properties of the RL algorithm driving this would be welcome.

Further, you discuss results from the field of automated mechanism design, but much of this work deals with mechanism families of provable guarantees (e.g. VCG and variants), and the setting here is different. Could you make the comparison a bit more detailed?
There are lot of resutls on collusion for vcg mechanisms (including based on cooperative game theory - e.g. the core and Shapley value). Can you say (or conjecture) anything along these lines here? In the case of this tacit collusion, the consumer loses welfare, but which of the sellers makes most of the gains?

All in all, this is a very interesting topic, and I like the approach presented.

---

> ### Author Response · Authors · 2022-08-01
> **Reviewer 7ViN Rebuttal**
>
> Thank you for your thorough feedback!
>
> Regarding weaknesses:
>
> Using Q learning for the pricing agents is a deliberate choice as we want to model the followers' behavior as previously done by Calvano et al. (2020), and this, together with the follow-on work of Johnson et al. (2020), gives us a clear baseline for our work. This noted, we are also testing alternate adjustment algorithms for pricing agents in parallel research projects. We have confirmed that collusive behaviors in the setting studied by Calvano et al. (2020) also arise when pricing agents use DQN, PPO (with discrete prices), and DDPG (continuous prices).
>
> Regarding the platform RL algorithm: we did not test algorithms other than A2C. However, from the previous Brero et al. (2021a) work, which considered Stackelberg learning in the context of mechanism design, we know that PPO is also successful in learning optimal policies for Stackelberg POMDPs.
>
> Regarding the sensitivity of our results.
>
> - In regard to the platform algorithm: we have only studied A2C with the default parameters and architectures provided by Stable Baselines3. There was no need for a hyperparameter search. The ablation studies focus instead on the methodologies introduced to solve the Stackelberg POMDP (see Appendix E).
>
> - In regard to the pricing agents: we deliberately use Q learning algorithms parameterized and initialized as in Johnson et al. (2020), in order to reproduce their results.
>
> Regarding theoretical results: Whenever our RL interventions find the optimal policy—which, in Figure 2 (right), happens more than 85% of the time —we converge to the behavioral Stackelberg equilibrium defined in (1). We will add these comments to our revision.
>
> Regarding properties of the RL algorithm: through StackPOMDP, our platform design problem reduces to finding an optimal policy in a single-agent POMDP, since the adaptation of the followers is absorbed into the environment. Thus, the properties you mention are those required of standard single-agent RL algorithms to derive optimal policies in POMDPs.
>
> Regarding the connection to mechanism design: this setting is very different from those studied in classical mechanism design, for example auction problems, in that this is a problem of moral hazard and not adverse selection (and we are screening sellers, who all have the same cost and thus have no private information, and our problem is not one of allocating resources to bidders). There is no technical connection to the literature on concerns around collusion in allocation mechanisms such as VCG.
>
> Regarding your questions: Testing additional RL algorithms, on the part of the pricing agents, would require a refactoring of our code base as we implemented multi-agent Q learning following the initializations and convergence criteria provided by Johnson et al. (2020). We think it is an advantage of our work that we are closely following previous work in the economics literature in all regards except for exploring the use of RL methods also to learn to mitigate collusion. This said we plan to explore interactions between an adaptive platform and pricing agents under alternate seller behaviors in future work.
>
> Regarding how the welfare is distributed among the sellers: the set-up is symmetric, with no a priori distinction between sellers, as they are Q-learners with the same costs and quality indexes. Even when considering each instance, under “No Intervention,” “State-based RL,” and “No State RL” sellers almost always split welfare equally (in some scenarios, one of the two sellers quotes a slightly higher price, achieving around 40% of the overall welfare). On the other hand, DPDP and PDP are asymmetric by design, as they display only one seller who achieves all the welfare.

---

### Official Review · Reviewer_VP8z · 2022-07-21

**Rating:** 4
**Confidence:** 3
**Soundness:** 3 good
**Presentation:** 2 fair
**Contribution:** 2 fair

**Summary:**

This paper aims to prevent collusive pricing by sellers through a learned framework that solves the two-level, Stackelberg problem. Specifically, the paper introduces the class of threshold policies that optimize consumer surplus and a method to learn optimal leader policies in this class. Then it is experimentally shown that the learned interventions outperform the hand-designed interventions proposed in earlier work.

**Questions:**

- I think maybe I have missed something, but it isn't super clear to me what do you mean by "collusion" or "collusive pricing" formally. Empirically, it seems the goal for the proposed methods is to maximize consumer surplus. Are these two objectives interchangeable? If not, please elaborate; if yes, then maybe the writing can be revised a bit.

- The title "Learning to Mitigate AI Collusion on Economic Platforms" sounds more general than the text actually addresses, which is limited to price collusion on e-commerce sites.

- The key contribution of this paper is using RL, for the first time, to prevent collusive pricing. Therefore, there is no prior RL work to compare to, but hand-crafted rules proposed in earlier work. Looking at Figure 2, the RL approaches start to outperform only after 1e7 training steps. This corresponds to a huge number of transactions, during which the seller's cost is assumed constant. How realistic is it? Shouldn't the cost be modeled rather than a constant? (For example to reflect the economy of scale. This point is also raised in the conclusion.)

**Limitations:**

N.A.

**Strengths And Weaknesses:**

Strengths:

- As algorithmic pricing may indeed facilitate collusive behaviors (though sometimes unintentionally), this is a super interesting research direction, especially considering the rising popularity of algorithmic pricing in the e-commence space.

- To cast this problem as multi-agent learning and model it as Stackelberg game is novel and interesting.

Weaknesses:

- This work is, to a great extent, a thought experiment with limited support in a real-world setting. Learning in an e-commence environment often involves noisy training data, unexpected user behaviors. Without real-world datasets/applications, it is hard to tell if the setups outlined in Section 5 are reasonable and useful.

---

> ### Author Response · Authors · 2022-08-01
> **Reviewer VP8z Rebuttal**
>
> Thank you for your thorough feedback!
>
> Regarding weaknesses: We chose to investigate the setting studied by Calvano et al. (2020), which has become the baseline model in the economics literature and is the setting where AI collusion has been studied. We agree it would be interesting to explore additional settings in future work, but for this first paper we prefer to focus on the Calvano et al. (2020) setting (and the use of rules to mitigate collusion, as studied by Johnson et al. (2020)). This gives a clear baseline against which to compare.
>
> Regarding your questions:
>
> - Following Calvano et al. (2020), by "collusion" we mean that (1) the prices quoted by the RL agents after convergence are above the Bertrand-Nash ones, and (2) the final RL strategies are such that deviations from the collusive agreements are punished by the other RL agents via lower prices. We will make all this precise in revising the paper. Collusion causes lower consumer surplus, this coming from higher prices, and consumer surplus is the measure of economic efficiency that the economic platform cares about.  In this regard, we follow Johnson et al. (2020) in adopting consumer surplus as the target of intervention.
>
> - On the title: Would "Learning to Mitigate AI Collusion on e-Commerce Platforms" address your concerns?
>
> - Similar to the point above, we use constant seller costs in line with Calvano et al. (2020). We are interested in studying other economic models, as well as alternate seller behavioral models, but wanted a clean baseline for this paper. For example, related work in economics has studied collusion in the context of energy supply (Abada and Lambin, 2022), and this is a suitable target for future work.
>
> - Regarding the high number of transactions: the reason for the large number of training steps is to allow the Q learning agents to re-equilibrate in response to a change in platform policy. For an  “in the wild” deployment, we would first train the rules of the platform in an offline environment to calibrate them before deployment and further adaptation. Faster adaptation by followers can also be achieved through a query-based seller model that directly attains an approximation to the outcome of Q-learning. We’ll add this discussion to our revision.

---

### Author Response · Authors · 2022-08-01
**New Figure 4 and Figure 7**

In the revised paper, we have updated two figures in the appendix (fixing a problem with the computational pipeline). Specifically,

- in Figure 4, both RL baselines reach optimal performance, with consumer surplus equal to 0.8;
- in Figure 7, State-based RL outperforms the DPDP after 15M training steps and not 45M, as reported.

These changes do not materially affect (but actually reinforce) the results or conclusions in the paper, in that our Stackelberg POMDP methodologies learn optimal interventions for fixed costs (and our state-based intervention is robust to out-of-distribution costs).

---

### Meta-Review · Area_Chair_Y35B · 2022-08-22

**Recommendation:** Accept
**Confidence:** Less certain

**Metareview:**

This paper proposes an RL based method to prevent collusive pricing by sellers through a framework that solves the two-level, Stackelberg problem.

The paper received a mixed evaluation from the reviewers, ranging from accept (7) to weak reject (4).

The strengths of the paper mentioned by the reviewers were:
- The considered problem was appreciated and acknowledged to be an important research direction
- Novelty in casting the problem as a Stackelberg game
- The proposed approach is competitive and robust

On the other hand, the identified weaknesses were:
- Potentially limited support in a real-world setting and a limited scope (only designed and testified in a specific economic model)
- Problems with the presentation

Despite the weaknesses mentioned above, I lean toward acceptance with my recommendation.

**Award:**

No

---

### Decision · Program_Chairs · 2022-09-14

Accept